# Development of Multiple-Heading-Date *mtl* Haploid Inducer Lines in Rice

**Jian Wang, Yuexuan Cao, Kejian Wang** and **Chaolei Liu** *

State Key Laboratory of Rice Biology, China National Rice Research Institute, Hangzhou 310006, China; 82101202227@caas.cn (J.W.); 82101196031@caas.cn (Y.C.); wangkejian@caas.cn (K.W.)
* Correspondence: liuchaolei@caas.cn

**Abstract:** In vivo doubled haploid (DH) production based on crossing heterozygous germplasm with *mtl* haploid inducer lines promises to transform modern rice (*Oryza sativa*) breeding. However, this technology is limited, as haploid inducers and pollen acceptors have asynchronous heading dates. To address this obstacle, we developed a panel of multiple-heading-date *mtl* haploid inducer lines that produce pollen for more than 35 days. We edited the *MTL* gene in a hybrid rice with the CRISPR-Cas9 system. We then selected transgene-free homozygous mutants in the $T_1$ generation and reproduced to $T_4$ generation by single-seed descent method. We obtained 547 *mtl* haploid inducers with diverse heading dates (from 73 to 110 days) and selected 16 lines comprising a core population with continuous flowering. The seed-setting rate and haploid induction rate (HIR) of the core panel were 4.0–12.7% and 2.8–12.0%, respectively. Thus, our strategy of using multiple-heading-date *mtl* haploid inducers could accelerate the use of in vivo DH technology in rice breeding.

**Keywords:** haploid inducers; *MTL*; multiple heading dates; rice; breeding; DH technology





## 1. Introduction

Generation of pure lines is an important component of crop breeding programs [1]. Breeders usually develop such breeding materials via 6–10 generations of recurrent selfing and selection after crossing [2]. However, this process is time-consuming and laborious, and the final inbred products still have residual heterozygosity, which might delay variety registration due to the distinctness, uniformity and stability criteria [3]. Doubled haploid (DH) technology provides a rapid alternative for producing pure lines, allowing completely homozygous lines to be reproduced in a single year. Besides this advantage, DH lines offer opportunities for increasing genetic gain [4]. Compared to the former conventional approach, DH technology shows great promise for crop breeding.

Haploid induction is the critical step of DH technology. Both in vivo (inter- and intraspecific hybridization; centromere-mediated haploidization) and in vitro (culture of immature male or female gametophytes) methods are available to generate haploids. The in vivo technology of haploid production has more advantages in breeding development compared with in vitro methods, which can avoid the waste of labor force and resources during haploid induction. In vivo haploid induction has become a hot spot in the development of DH breeding. However, there are still many problems to be solved with respect to this technology [5,6]. It is difficult to produce enough DH lines for breeding programs by using in vitro methods [3,7]. For example, the anther culture, a common method used for haploid production in crops, is restricted by genetic factors of different rice varieties [8]. In contrast, haploidization via intraspecific hybridization using haploid inducers has been proven to be efficient for the large-scale production of DH lines and is widely used in maize (*Zea mays*) breeding programs [3,4]. To date, hundreds of commercial maize varieties have been developed based on this method [3]. However, maize is the only successful crop to be bred with DH technology because of the development of high-HIR haploid inducers.

The genetic basis of high-haploid induction in maize has recently been elucidated. Prigge et al. [2] identified two major quantitative traits loci (QTLs), *qhir1* and *qhir8*, for haploid induction rate (HIR), which explain 66% and 20% of the genotypic variation in this trait, respectively. Later, *qhir1* was cloned as *ZmMTL* (also named as *NLD* or *ZmPLA1*), which encodes a pollen-specific phospholipase. Loss of function of this gene triggers haploid induction in maize [9–11]. *MTL* is conserved in cereals, and genome-edited lines of its homologous genes in rice and wheat (*Triticum aestivum*) can also be used as haploid inducers [7,12–14]. *qhir8* was ultimately identified as *ZmDMP* by map-based cloning [15]. *ZmDMP* acts as an enhancer of haploid induction, and it significantly increases HIR (by twofold to threefold) in the presence of *ZmMTL* [15]. However, *ZmDMP* has no highly homologous gene in rice.

The successful development of a haploid induction system by genome editing of *OsMTL* makes DH technology promising in rice breeding programs [7,12]. However, this emerging system in rice has several barriers, such as the low HIR and seed set of haploid inducers, the inefficiency of haploid identification, chromosomal doubling of rice haploids, etc. [7]. Here, we report the development of multiple-heading-date *mtl* haploid inducers that can solve the problem of asynchronous heading dates between haploid inducers and pollen acceptors. The pollen from the 16 investigated *mtl* haploid inducers is available for more than 35 days, making it possible to perform field crossing for an extended period.

## 2. Materials and Methods

### 2.1. CRISPR/Cas9 Vector Construction and Plant Transformation

Two target sites, labeled as BJF and BJG, were selected within the *OsMTL* gene, and their targeting specificity was confirmed according to the method described by Hsu et al. [16]. The double-stranded overhangs of $MTL^{BJF}$ and $MTL^{BJG}$ oligonucleotides (Supplementary Table S1) were separately ligated into the SK-sgRNA vectors digested with AarI. Subsequently, the sgRNAs for $MTL^{BJF}$ (digested with KpnI/BglII) and $MTL^{BJG}$ (digested with KpnI/BglII) were assembled into *pC1300-Actin*:Cas9 binary vectors (digested with KpnI/BamHI) using $T_4$ ligase. The final constructed vectors were named *pC1300-Actin*:Cas9-sgRNA$^{BJF}$ and *pC1300-Actin*:Cas9-sgRNA$^{BJG}$.

The *indica-japonica* hybrid rice variety 'Chunyou 84' (CY84) was used for genetic transformation. The constructed vectors, *pC1300-Actin*:Cas9-sgRNA$^{BJF}$ and *pC1300-Actin*:Cas9-sgRNA$^{BJG}$, were separately introduced into CY84 via the *Agrobacterium*-mediated transformation (strain EHA105) method described by Hangzhou Biogle Co., Ltd. (Hangzhou, China). The $T_0$ transgene plants were grown in a greenhouse maintained at average day and night temperatures of 30 °C and 25 °C, respectively, with 75% relative humidity.

### 2.2. Detection of Mutations

Approximately 100 mg of young leaf tissue of transgenic plants was ground into a power by a tissuelyser (Jingxin, Shanghai, China). Genomic DNA was extracted from the samples using the CTAB method. KOD FX DNA polymerase (Toyobo, Osaka, Japan) was used to amplify the fragments surrounding the two target sites. The PCR products were sequenced by the Sanger method and analyzed by the degenerate sequence decoding method [17]. Primers used for amplification are listed in Supplementary Table S1.

### 2.3. Field Experiments

The *mtl* haploid inducer plants were grown annually at the experimental farms of China National Rice Research Institute in Hangzhou (119°54′ E, 30°04′ N) and Lingshui (110°00′ E, 18°31′ N), China. They were reproduced to the next generation by single-seed descent method. Paddy water management, fertilization and crop protection followed the local farming practices.

To measure heading date, seeds of the *mtl* haploid inducers ($T_4$ generation) were soaked in water for two days and sown in a seed bed in Hangzhou on 30 May. Twenty-five-day-old seedlings of each line were transplanted into a four-row plot with six plants per

row. The heading date was defined as the date when more than 50% of individuals had reached the heading stage.

### 2.4. HIR Measurement

The haploid inducers (used as the male parent) were crossed with ZhongguangA (used as the female parent) to investigate their HIR. The female parent was a cytoplasmic male sterile line commonly used in three-line hybrid rice breeding. The hybrid seeds were harvested and germinated in the laboratory, and the seedlings were cultured in a hydroponic solution as described by Liu et al. [18]. Genomic DNA was extracted from these plants, and their BJF or BJG target sites were genotyped by Hi-TOM technology [19]. The homozygote lines without mutation were classified as haploids or double haploids. The Hi-TOM primers used for amplification are shown in Supplementary Table S1.

### 2.5. DNA Ploidy Analysis

Approximately 2 cm$^2$ of fresh leaf tissue was used for DNA ploidy analysis. Samples were prepared according to a previously described method [12]. The ploidy of leaf cells was determined by estimating the nuclear DNA content using flow cytometry. All procedures were performed at 4 °C or on ice.

## 3. Results

### 3.1. Strategy for Obtaining Asynchronous Flowering Time

Seed-based haploid induction based on *mtl* is a promising approach for rice breeding, but large-scale hybridizations are required to produce enough DH lines for phenotypic selection [7]. When performing crosses, it is important to choose haploid inducers with a suitable flowering time for the pollen-accepted materials. To address this problem, we employed a strategy that allows pollen to be available for a long period of time by generating multiple-heading-date *mtl* haploid inducers. A brief flow chart was drawn before starting the work (Figure 1). We planned to knock out the *OsMTL* gene in hybrid rice using the CRISPR-Cas9 system. The goal was to produce offspring with diverse flowering times due to the separation and recombination of genetic material. In the $T_0$ and $T_1$ generations, homozygous *mtl* mutant lines were screened, and their exogenous components were be removed. Then, the selected lines were reproduced to advanced generation by single-seed descent method. Finally, multiple-heading-date *mtl* haploid inducers were developed, and their HIR was detected.

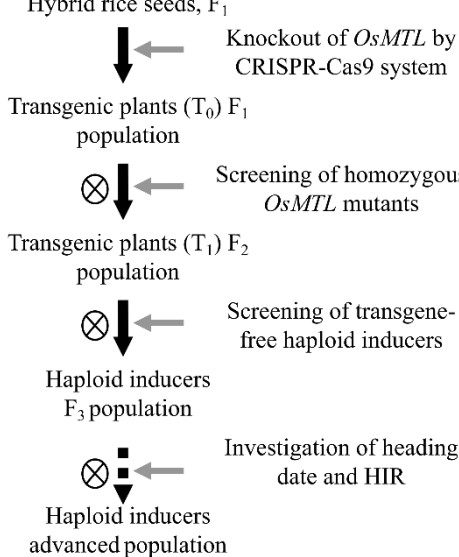

**Figure 1.** Flow chart of the method used to develop multiple-heading-date *mtl* haploid inducer lines in rice.

### 3.2. Knockout of OsMTL in Chunyou84

To generate *mtl* haploid inducers, we constructed vectors targeting the *OsMTL* gene (Figure 2a) with the CRISPR/Cas9 genome editing system [20]. The guide RNAs targeted BJF and BJG sites located in exon 1 and exon 4 of *OsMTL*, respectively (Figure 2b). We chose Chunyou84 (CY84) for transformation, as the progeny of this typical *indica-japonica* hybrid rice variety segregate into lines with widely variable heading dates [21]. Four homozygous mutants were obtained in the $T_0$ generation: lines #1 (4-bp deletion) and #2 (1-bp insertion) at the BJF site and lines #3 (1-bp deletion) and #4 (1-bp insertion) at the BJG site (Figure 2b). These mutant lines were grown in a greenhouse, and their seeds were harvested. We investigated the HIR of these haploid inducers and found that the HIR value showed no significant differences between BJF- (10/238 = 4.2%) and BJG- (8/194 = 4.1%) targeted lines.

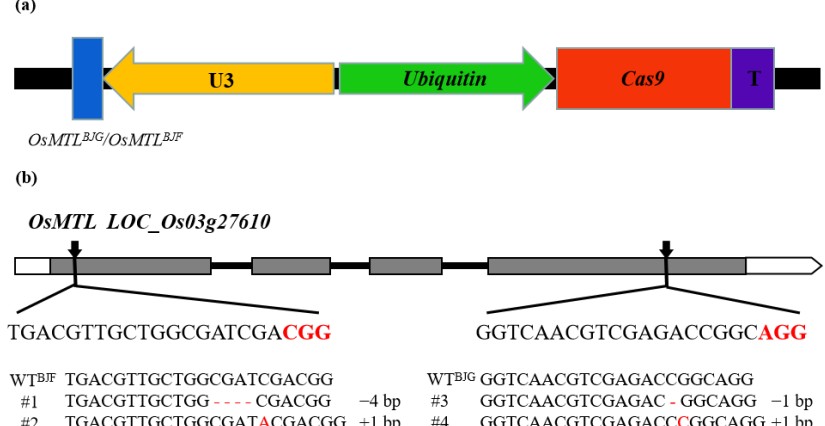

**Figure 2.** Knockout of *OsMTL* in Chunyou84 by the CRISPR-Cas9 system. (**a**) Schematic diagram of the structure of the CRISPR-Cas9 vector targeting *OsMTL*. (**b**) Targeted sites in *OsMTL* gene and mutations at the BJF and BJG loci in $T_0$ generation plants. Mutations with 1 bp insertions are indicated by red letters, and the deleted sequences are indicated by red hyphens.

In the $T_1$ generation, we used two pairs of primers, Hyg F/R and Cas9 F/R, that target hygromycin and Cas9, respectively, to screen transgene-free lines. We obtained 22 independent lines of each type of mutant line, the PCR products of which were amplified using the above two primers; lines that generated no simultaneous electrophoretic band were considered transgene-free lines. A total of 16 lines were confirmed to have no exogenous T-DNA elements: five lines for line #1, two lines for line #2, two lines for line #3 and seven lines for line #4 (Figure 3). We used these transgene-free lines to generate offspring.

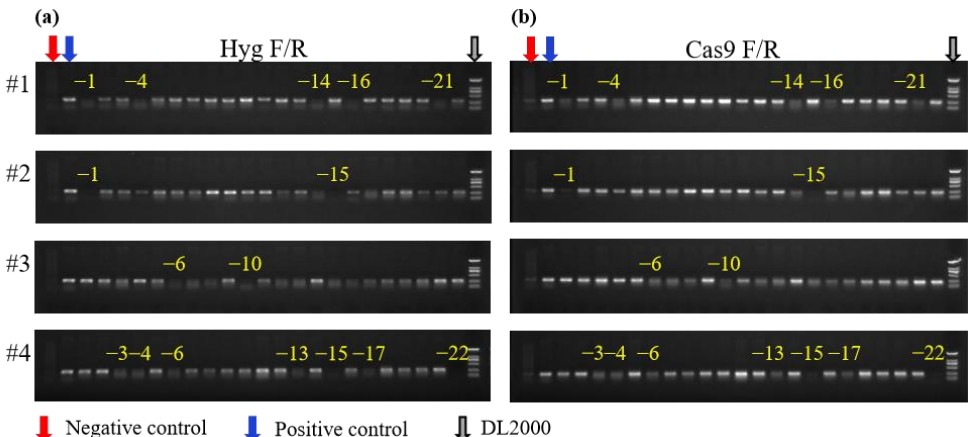

**Figure 3.** Screening of transgene-free haploid inducers. Genomic DNA from 22 $T_1$ progeny plants of lines #1, #2, #3 and #4 was amplified with primers Hyg F/R (**a**) and Cas9 F/R (**b**). Transgene-free haploid inducers are indicated by yellow numbers.

### 3.3. Generation of mtl Haploid Inducers with Multiple Heading Dates

To date, we have reproduced the *mtl* haploid inducers to the $T_4$ generation via self-fertilization. We generated 547 haploid inducers, the heading dates of which varied from 73 to 110 days (Figure 4a). However, this population was too large, as intensive labor and field resources would be required to maintain it each year. As the flowering time of a single line could last for 3–5 days, we selected two lines every five days based on heading date time to compose a minimum continuous flowering population. Ultimately, we obtained a panel of 16 *mtl* haploid inducers, the pollen of which was available for more than 35 days (Figure 4b).

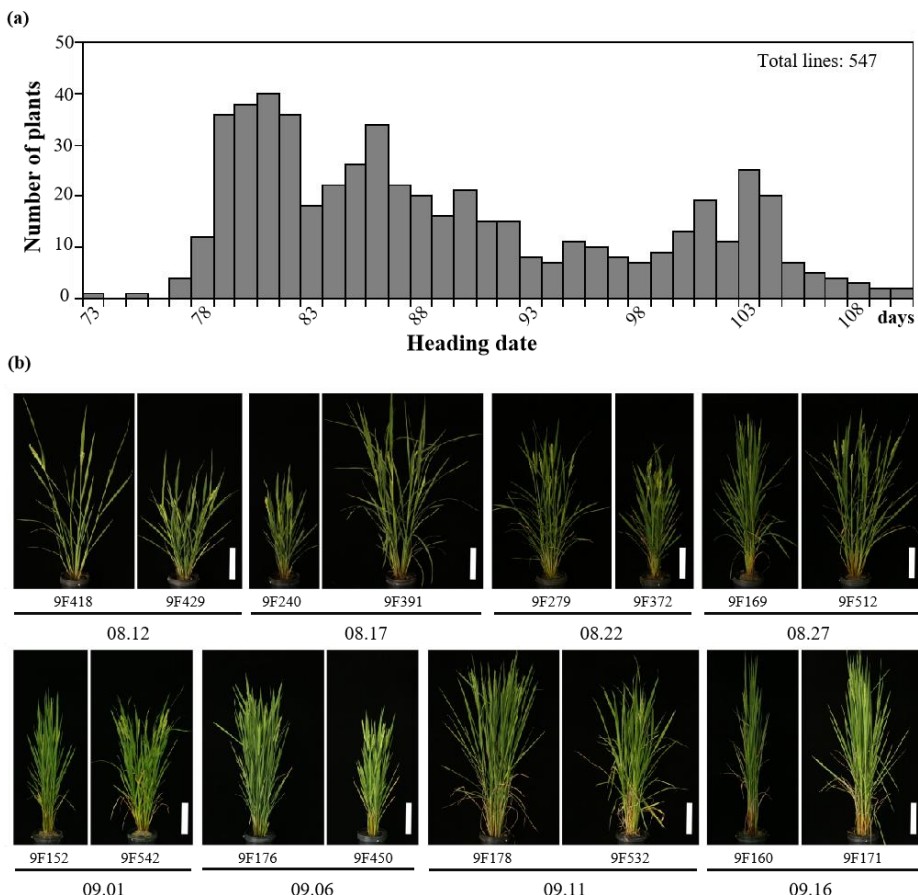

**Figure 4.** Multiple-heading-date *mtl* haploid inducers. (**a**) Heading date frequency distribution in the 547 RILs derived from transgene-free haploid inducer lines #1, #2, #3 and #4. (**b**) Plant morphology and heading date of 16 *mtl* haploid inducers. Seeds were sown on 30 May, and plants were grown in Hangzhou, China. Bar = 20 cm.

HIR is an important trait of inducers that is related to the proportion of haploid seeds obtained in induction crosses [22]. To rapidly investigate the HIR of the selected haploid inducers, we crossed cytoplasmic male sterile lines with the haploid inducers due to the convenience of field crossing and the lack of self-setting seeds. In the hybrid seeds, we detected the *MTL* target mutation sites to screen haploids (or double haploids). Similar to previous reports [7,12], the induced rice haploid plants were smaller than the diploid plants, with reduced plant height, as well as decreased panicle length and glume size (Figure 5). The HIR of these 16 *mtl* haploid inducers varied from 2.8 to 12.0%, with an average of 6.8% (Table 1). Their seed-setting rates were also investigated, which were 4.0~12.7% (Table 1). These results confirmed the inference that introducing *mtl* into diverse genetic backgrounds might yield inducers with a higher or lower HIR and seed-setting rate [7].

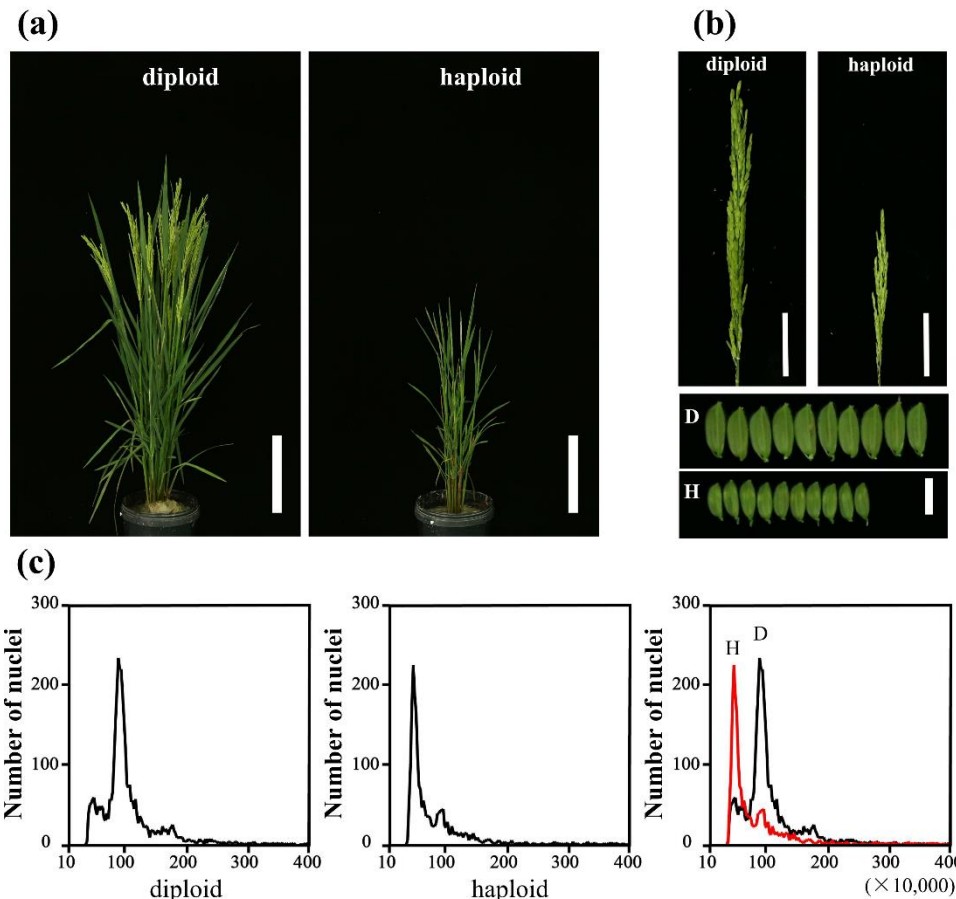

**Figure 5.** Characterization of a rice haploid line. (**a**) Whole plants of diploid and haploid rice. (**b**) Panicles and glumes of diploid and haploid rice. Bar = 20 cm (**a**), 3 cm ((**b**), upper panel) and 2 cm ((**b**), lower panel). (**c**) Ploidy analysis of diploid (D; black) and haploid (H; red) lines by flow cytometry. The haploid lines were generated by crossing ZhongguangA with *mtl* haploid inducers. Plants were grown in Hangzhou, China.

**Table 1.** Seed-setting rate and haploid induction rate of selected *mtl* haploid inducers in Hangzhou.

| Code | Seed-Setting Rate (%) | Hybrid Seeds [a] | Haploid + DH | Haploid Induction Rate (%) |
|------|----------------------|------------------|--------------|----------------------------|
| 9F418 | 8.6 ± 3.8 | 158 | 10 | 6.3 |
| 9F429 | 6.6 ± 1.8 | 138 | 9 | 6.5 |
| 9F240 | 4.5 ± 2.8 | 125 | 15 | 12.0 |
| 9F391 | 8.8 ± 1.4 | 117 | 4 | 3.4 |
| 9F279 | 12.7 ± 2.9 | 144 | 4 | 2.8 |
| 9F372 | 4.3 ± 0.8 | 139 | 8 | 5.8 |
| 9F169 | 7.7 ± 2.9 | 106 | 10 | 9.4 |
| 9F512 | 4.7 ± 0.6 | 92 | 6 | 6.5 |
| 9F152 | 9.0 ± 1.6 | 128 | 6 | 4.7 |
| 9F542 | 8.7 ± 0.7 | 120 | 7 | 5.8 |
| 9F176 | 6.2 ± 0.5 | 99 | 5 | 5.1 |
| 9F450 | 11.4 ± 1.0 | 142 | 11 | 7.7 |
| 9F178 | 4.0 ± 0.9 | 111 | 12 | 10.8 |
| 9F532 | 9.4 ± 3.1 | 103 | 5 | 4.9 |
| 9F160 | 10.0 ± 1.4 | 154 | 13 | 8.4 |
| 9F171 | 9.5 ± 0.5 | 123 | 11 | 8.9 |

Note: [a] Hybrid seeds were a cross between *mtl* haploid inducers (used as male parent) and ZhongguangA (used as female parent).

The relationship between HIR and heading date or seed-setting rate in rice remains unclear. Thus, performed a correlation analysis of these traits using the 16 *mtl* haploid inducers. We found that HIR exhibited a weak correlation with heading date, with a

correlation coefficient of 0.22 ($p$ = 0.41) (Figure 6a). In contrast, a highly negative correlation was observed between HIR and seed-setting rate (r = −0.45, $p$ = 0.08) (Figure 6b).

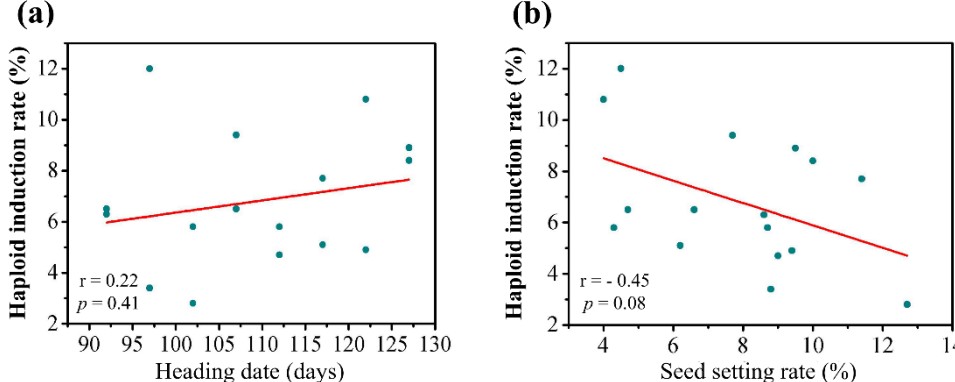

**Figure 6.** Correlation analysis between haploid induction rate (HIR) and heading date (**a**) and between HIR and seed-setting rate (**b**).

## 4. Discussion

### 4.1. Approaches for Solving the Problem of Asynchronous Heading Date

The discovery of haploids in higher plants led to the use of DH technology in plant breeding. DH can accelerate plant breeding and genetic research based on the 100% homozygosity genotype after haploid induction in one generation. Efficient haploid induction is the first barrier. Knockout *MTL* genes can cause rice to induce haploids. However, its 2–6% hybridization induction rate limits the acquisition of a large number of haploid seeds [7], which requires a large number of hybridizations. In this study, we selected sterile lines as pollen recipients to make field work more efficient. However, the problem of asynchronous heading date must be solved when a large number of cross combinations are needed. Several approaches can be considered to solve this problem. The solution of sowing at different dates is commonly used to arrive at the same flowering date with different materials [23]. We sowed *mtl* haploid inducer seeds at different times to generate inducers with different flowering dates. However, the *mtl* mutants never have enough seeds because of the low seed-setting rate [12] (Table 1). In addition, different sowing dates mean tedious work is required in the field. Progenies of hybrid rice exhibit various phenotypes due to the segregation and recombination of genetic material [24]. Therefore, in this study, we introduced *mtl* in hybrid rice to generate haploid inducers with multiple heading dates to solve the problem of synchronous heading dates.

CY84 is a typical *indica-japonica* hybrid rice variety [21]; thus, its progeny, which are derived from self-fertilization, show large variation in heading date. Here, we knocked out the *OsMTL* gene in the CY84 variety and advanced the homozygous mutants for three generations. We obtained 547 *mtl* haploid inducers with heading dates ranging from 73 to 110 days (Figure 4a). These inducers provided pollen for more than one month, thereby solving the problem of asynchronous heading date. However, it is unnecessary to maintain such a large population only to provide inducers with pollen. Thus, we developed a minimal continuous flowering population composed of 16 *mtl* lines (Figure 4b). This minimal population not only provides inducers with pollen for an extended period but also simplifies the field work. In addition, correlation analysis revealed a low correlation between HIR and heading date (Figure 6a), indicating that heading date has no significant effects on HIR.

### 4.2. Advantages of the mtl Haploid Inducer Population

Besides solving the problem of asynchronous heading dates, the *mtl* haploid inducer population has other possible uses for either applied or basic research. To produce large numbers of DH lines, it is necessary to generate an inducer with an acceptable HIR (usually ≥6%) [3]. Yao et al. [7] proposed introducing *mtl* in diverse rice varieties to generate

inducers with a higher HIR. As our *mtl* haploid inducers were derived from a hybrid rice, CY84, they also exhibited variation in HIR traits (Table 1), indicating that other genetic factors, such as *ZmDMP*, affect HIR. Among the detected *mtl* inducers, two lines had HIR >10%, showing great promise in DH technology and are also likely to have higher HIR among the 547 *mtl* haploid inducers.

*OsMTL* is a component gene of the *Fix* strategy, which can fix heterosis in hybrid rice [12]. However, the low fertility resulting from the mutation of *OsMTL* limits the production of clonal seeds when using the *Fix* strategy [12,25]. Improvements in fertility are required to enable the technology to be commercialized [12]. According to our results, the *mtl* inducers had low seed set but showed variations in their genetic backgrounds (Table 1). Thus, other genetic factors must also be involved in this process. To identify the underling genetic mechanism for high fertility, as well as high HIR, we plan to sequence the *mtl* haploid inducer population and clone the candidate genes in the future.

## 5. Conclusions

Asynchronous heading dates between haploid inducers and pollen acceptors represent a troublesome barrier to DH technology. Here, we provided a convenient and rapid method to solve this problem by generating multiple-heading-date *mtl* haploid inducers in rice. Based on our strategy, the application of DH technology could be accelerated in rice breeding programs.

**Supplementary Materials:** The following are available online at https://www.mdpi.com/article/10.3390/agriculture12060806/s1, Table S1: Primers used in this study.

**Author Contributions:** K.W. and C.L. managed the project. J.W., C.L. and Y.C. performed the experiments. C.L. and J.W. analyzed the data. J.W. and C.L. wrote the manuscript. K.W. revised the manuscript. All authors have read and agreed to the published version of the manuscript.

**Funding:** This work was supported by the National Natural Science Foundation of China (32001527, 32188102 and 32025028) and the Earmarked Fund for China Agriculture Research System.

**Institutional Review Board Statement:** Not applicable.

**Informed Consent Statement:** Not applicable.

**Data Availability Statement:** All datasets used in this study are included in the manuscript and the supplementary file.

**Conflicts of Interest:** The authors declare that the research was conducted in the absence of any commercial or financial relationships that could be construed as a potential conflict of interest.

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
