# Peer review of "Development of Multiple-Heading-Date mtl Haploid Inducer Lines in Rice"

_agriculture, doi:10.3390/agriculture12060806_

Round 1
Reviewer 1 Report
Revision of the --Manuscript Draft-- Manuscript Number: agriculture-1731897
Title: Development of multiple heading date mtl haploid inducer lines in rice
Type: Article
This is a manuscript within the scope of the journal "Agriculture (ISSN 2077-0472)" and presented in a well-structured manner. It is relevant for providing a convenient and rapid method to solve the problem via generating multiple heading date mtl haploid inducers in rice breeding program.
The actual manuscript needs to be corrected to minor text editing. I suggest the authors for improving the ms to take care for the whole document e.g. LNs 15, 46, 105, 106, 175. In the Figures section, please, see the LN 168.
With regards
Author Response
Point 1: The actual manuscript needs to be corrected to minor text editing. I suggest the authors for improving the ms to take care for the whole document e.g. LNs 15, 46, 105, 106, 175. In the Figures section, please, see the LN 168.
Response 1: Thanks for your careful review on our manuscript and your kind comments. We have checked the mentioned points as well as other mistakes throughout the full manuscript carefully, and corrected them in red.

Reviewer 2 Report
Review comments on “Development of multiple heading date mtl haploid inducer lines in rice”
General comments
This is general research on improving doubled haploid (DH) production based on crossing heterozygous germplasm with mtl haploid inducer lines promises to transform the modern rice (Oryza sativa) breeding, Authors applied the MTL gene in hybrid rice by CRISPR-Cas9 system and got early flowering rice. I think the paper is well organized, but the authors should do a major revision to improve the manuscript quality. The detailed comments are listed below,
1. BJF and BJG were selected within the OsMTL gene, please describe their functions in the Introduction;
2. What’s the function of the target genes in the expression?
3. Where is the knockout genes? And did you test the expression in the F2?
4. Please use high resolution images, it is not clear to see the breeding;
5. Is there any grain quality after gene edition? It is interesting for the readers.
Author Response
Point 1: BJF and BJG were selected within the OsMTL gene, please describe their functions in the Introduction;
Response 1: Thank you very much for your review. BJF and BJG are the two different knockout target sites, which are located in the exon 1 and exon 4 of OsMTL, respectively. Actually, we started this research in 2017. At that time, we thought the target sites might influence the HIR (Haploid Induction Rate) of the haploid inducers. However, the HIR of BJF and BJG mtl mutants in CY84 variety (T0) were 4.2% and 4.1%, respectively, indicating that the different knockout target sites in the OsMTL gene had no significant effects on HIR. Yao et al. also desigened two target sites in OsMTL gene and obtained similar results (Yao et al., OsMATL mutation induces haploid seed formation in indica rice. Nature Plants, 2018). Therefore, BJF and BJG are just different knockout target sites within the OsMTL gene here. We have described them in detail in Line 138-148 as well.
Point 2: What’s the function of the target genes in the expression?
Response 2: The function of the target gene MTL has been extensively studied in crop plants, and mutation of MTL can trigger haploid induction (Kelliher et al., MATRILINEAL, a sperm-specific phospholipase, triggers maize haploid induction. Nature, 2017; Yao et al., OsMATL mutation induces haploid seed formation in indica rice. Nature Plants, 2018; Liu et al., Efficient induction of haploid plants in wheat by editing of TaMTL using an optimized Agrobacterium-mediated CRISPR system. J Exp Bot, 2020; and so on). The MTL gene was specially expressed in pollen, which has been well studied in several researches (Kelliher et al., MATRILINEAL, a sperm-specific phospholipase, triggers maize haploid induction. Nature, 2017; Jiang et al., A reactive oxygen species burst causes haploid induction in maize. Mol Plant, 2022; Sun et al., Matrilineal empowers wheat pollen with haploid induction potency by triggering postmitosis reactive oxygen species activity. New Phytol, 2022). We have introduced the function and expression information of MTL gene in Line 51-56.
Point 3: Where is the knockout genes? And did you test the expression in the F2?
Response 3: Thank you for your question. The knockout gene is OsMTL (Os03g0393900, or LOC_Os03g27610) in our study, which is located on the Chromosome 3 in rice genome. As knockout of OsMTL can induce haploids, we focused on the mutation type of OsMTL genome editing lines (Figure 2). We also have checked the target sites genotype in the F2 population, and found the same mutation type with that of the F1 lines (CY84 T0 ) (Figure 2). We have not tested the expression of the edited OsMTL gene in F2 because that the expression of osmtl was not important for haploid induction here.
Point 4: Please use high resolution images, it is not clear to see the breeding;
Response 4: Thank you for your suggestion. We have uptaded high resolution images now.
Point 5: Is there any grain quality after gene edition? It is interesting for the readers.
Response 5: Many thanks. Here, the osmtl knockout lines were used as haploid inducers in the DH breeding technology. In this DH technology, the pollen from haploid inducers (used as male parent) were pollinated to heterozygous materials (used as female parent). In their F1 progeny, 2.8~12.0% female-derived haploids were generated, with no genetic information gained from haploid inducers. More in detail, the whole genome of haploid inducers is eliminated during haploid formation. This means that the grain quality of the desired haploids determined by the genetic background of female parent, with no relationship with that of haploid inducers. Thus, the grain quality of osmtl haploid inducers is always neglected in this research area. In this study, we have not tested rice quality as well. Anyway, we think that the grain quality is not effected after gene edition of OsMTL gene in theory.

This manuscript is a resubmission of an earlier submission. The following is a list of the peer review reports and author responses from that submission.
Round 1
Reviewer 1 Report
The research paper submitted to the journal fit partially within the general scope of the Journal. The paper should be submitted to a more specific Journal dealing with breeding from the MDPI journals.
I have several major concerns related to the originality of the data which seems quite low. The data set is very limited and this could not justify publication. The whole paper has a total of 5 pages which cannot publish in prestigious journals such as Agriculture MDPI. In fact, the discussion section is very descriptive. The authors should know that it is not enough in the discussion section to report if their results are in line or not with the previously published research but why. The field validation was conducted for only one year which is not enough to have solid conclusions. Based on these considerations the paper should not be accepted for publication in Agriculture MDPI.
Reviewer 2 Report
Revision of the --Manuscript Draft-- Manuscript Number: agriculture-1617880
Title: Development of multiple heading date mtl haploid inducers in rice
Type: Article
This is a manuscript within the scope of the journal "Agriculture". The main aspects of my revision are related with corrections to minor methodological errors and text editing. They are mentioned in the revised and attached document.
With regards

Reviewer 3 Report
Dear authors,
Manuscript Number: Agriculture -1617880: Development of multiple heading date mtl haploid inducers in rice.
The manuscript is well organized and written, I have only the suggestion to include and comment a very recent review on the same topic that is reported below:
Kyum, M., Kaur, H., Kamboj, A., Goyal, L., & Bhatia, D. (2022). Strategies and prospects of haploid induction in rice (Oryza sativa). Plant Breeding, 141( 1), 1– 11. https://doi.org/10.1111/pbr.12971
Kind Regards
